



# Are tidal predictions a good guide to future extremes? - a critique of the Witness King Tides Project

John Hunter[1]

[1]Institute for Marine and Antarctic Studies, University of Tasmania, Hobart, Tasmania, Australia

**Correspondence:** John Hunter (jrh@johnroberthunter.org)

**Abstract.** An analysis of the viability of the *Witness King Tides Project* (hereafter called *WKT*) using data from the GESLA-2 database of quasi-global tide-gauge records is described. The results indicate regions of the world where *WKT* should perform well (e.g. the west coast of the USA) and others where it would not (e.g. the east coast of North America). Recommendations are made both for assessments that should be made prior to a *WKT* project, and also for an alternative to *WKT* projects.

## 1 Introduction

This work was originally stimulated by the *Witness King Tides Project* (hereafter called *WKT*), which originated in New South Wales, Australia (Watson and Frazer, 2009), and is now internationally active in a number of regions, especially the USA and Australia (King Tides Project: http://www.kingtides.net, accessed: 20 November 2019). *WKT* is a citizen-science project
10 designed to collect photos of the shoreline at the time of annual highest astronomical tide, with the aim of indicating the flooding that may occur routinely with sea-level rise. Participants are informed of the annual highest astronomical tide in their region for a given year and are asked to photograph their local shoreline at this time (hereafter called a *WKT Day*).

A critical assumption of *WKT* is that the annual highest astronomical tide is a good proxy for the actual highest water level during the year, both in timing and height. There are two potential problems with this approach: (a) that the water level on the
15 *WKT Day* may be significantly modified by a storm surge and (b) a significantly higher water level may occur at a different time of the year from the *WKT Day* due to the coincidence of a large positive surge and an astronomical tide that is lower than the one on the *WKT Day* (so the opportunity of getting more dramatic photos at this alternative time is lost). Regarding (a), during the first *WKT Day* on 12 January 2009 in New South Wales, Australia, the observed maximum water level was 0.09 metres *below* the maximum astronomical tide, presumably due to a negative storm surge (Watson and Frazer, 2009). By way
20 of comparison, 0.09 metres is roughly the global-average sea-level rise from 1970 to 2009, raising the obvious question: 'how well is *WKT* likely to demonstrate the impact of future climate change if the photographed water level may be lower than expected by an amount equivalent to about 40 years of sea-level rise?'. A significant negative storm surge on a *WKT Day* may well give the unintended message that the impact of sea-level rise is likely to be unimportant.



This study uses the Global Extreme Sea Level Analysis Version 2 (GESLA-2) database of quasi-global 'high-frequency' (i.e. sampled at least hourly) tide-gauge records (Woodworth et al., 2017) to compare the statistics of annual maxima in the astronomical tide and in tidal observations. The results indicate how well *WKT* should work at over 300 locations around the world.

It should be noted that, for some locations and some years, there are more than one astronomical tides of similar magnitude to the maximum. In these cases, more than one *WKT Day* may be declared for that year. However, the analysis to be described here only considers the case of a single *WKT Day* during the year.

## 2  Methods

The GESLA-2 tide-gauge database contains 39,151 station-years of data from 1,355 stations (Woodworth et al., 2017). Most
of this data was sampled hourly and the remainder more frequently. GESLA-2 data is composed of two data sets, one denoted 'public' (which contains data for most of the world) and the other denoted 'private' (which mainly contains data for Australia). For the present analysis, these data sets were combined and were downloaded on 11 March 2016 (for the 'private' data) and 19 March 2016 (for the 'public' data). Individual years from the tide-gauge records were selected as follows:

1. observed heights that departed by more than 10 standard deviations from the average were rejected,

2. observed heights were binned to produce hourly values (this only affected the relatively few records that were sampled more frequently than hourly),

3. years with less than 80% of hourly values were rejected, and

4. years for which the two-year period centred on the the middle of the year had less than 80% of hourly values were rejected (this related to the tidal analysis – see later).

After this selection process, only tide-gauge records that contained at least 20 valid years were used for the results presented here. This represented a compromise between selecting long records and many records, and yielded data from 586 individual GESLA-2 *records*. Henceforth, a *record* (i.e. italicised) refers to an individual GESLA-2 record that contained at least 20 valid years. In some cases, more than one *record* occupied a given location. For example, data from the same location has sometimes been sourced from different data providers, in which case they generally cover different periods and are of different
lengths; such *records* are therefore, to a certain extent, independent and were analysed individually. Also, a significant number of *records* are from distinct, but relatively close, locations; for example, of the 171,405 separation distances between the 586 *records*, around 180 (0.1%) are less than 3 km. Consequently, for the maps produced in Figs. 1 to 6 and in Fig. 10, the results for some *records* would be obscured by the results for other nearby *records*. For this reason, the number of *records* was 'pruned' down from 586 to 311 using the 'neighbourhood' technique described in Appendix A. From each *neighbourhood*, the *record*
with the most years of data was selected for display in Figs. 1 to 6 and in Fig. 10. It should be stressed that this process involves





no averaging; it is simply a process of removing *records* that probably have less significant results (based on the fact that they are shorter) and that would otherwise obscure the results of their neighbours when plotted on a global map.

For each *record* (denoted by index $k$) and for each valid year (called here the *target year*; denoted by index $j$), the following analysis was performed:

1. A tidal analysis for 102 constituents was performed on the two-year period centred on the *target year*. A two-year analysis was performed because, for a few *records*, a one-year analysis failed using 102 constituents presumably because, for some constituent pairs, the Rayleigh criterion is only just satisfied. From this analysis, tidal predictions were performed for the times of all observations during the *target year*.

2. For each day, two periods were defined: a *civil day* (denoted by the subscript $c$), which is the full 24-hour day, and a
*daylight day* (denoted by the subscript $d$), which represents the period over which a natural-light photo may reasonably be taken and which is here (somewhat arbitrarily) defined as occupying 80% of the time between sunrise and sunset (therefore starting at 10% of the sunrise-to-sunset time after sunrise and ending at 10% of the sunrise-to-sunset time before sunset). Sunrise and sunset times were calculated using the *sunazimuth* program [1].

3. For each *record*, $k$, each valid year, $j$ and for each 'day', $i$, the following were calculated for both *civil days* and *daylight*
*days* (noting that, due to missing data, there are missing values of $i$ and $j$):

(a) the highest predicted tide for each 'day' (denoted $p_c(i,j,k)$ for *civil days* and denoted $p_d(i,j,k)$ for *daylight days*), and

(b) the highest observed sea level for each 'day' (denoted $o_c(i,j,k)$ for *civil days* and denoted $o_d(i,j,k)$ for *daylight days*).

4. For each *record*, $k$, and each valid year, $j$ the following were calculated for both *civil days* and *daylight days*:

(a) the day of the highest predicted tide during each valid year (denoted $I_{pc}(j,k)$ for *civil days* and denoted $I_{pd}(j,k)$ for *daylight days*). The highest predicted tide during each valid year is therefore given by $p_c(I_{pc}(j,k),j,k)$ for *civil days* and $p_d(I_{pd}(j,k),j,k)$ for *daylight days*.

(b) the day of the highest observed sea level during each valid year (denoted $I_{oc}(j,k)$ for *civil days* and denoted
$I_{od}(j,k)$ for *daylight days*). The highest observed sea level during each valid year is therefore given by $o_c(I_{oc}(j,k),j,k)$ for *civil days* and $o_d(I_{od}(j,k),j,k)$ for *daylight days*.

5. The following three *annual metrics* were obtained for each kind of 'day' and for each valid year:

(a) the *annual first metric*, which is the height of highest observed sea level above the observed maximum on the day of the highest predicted tide for the year, given by $o_c(I_{oc}(j,k),j,k) - o_c(I_{pc}(j,k),j,k)$ for *civil days* and
$o_d(I_{od}(j,k),j,k) - o_d(I_{pd}(j,k),j,k)$ for *daylight days*.

---

[1]https://sidstation.loudet.org/sunazimuth-en.xhtml





(b) the *annual second metric*, which is the number of days when the observed sea level ($o_c(i,j,k)$ for *civil days* and $o_d(i,j,k)$ for *daylight days*) was higher than the observed maximum on the day of the highest predicted tide for the year ($o_c(I_{pc}(j,k),j,k)$ for *civil days* and $o_d(I_{pd}(j,k),j,k)$ for *daylight days*), and

(c) the *annual third metric*, which is the height of the highest observed sea level on the day of the highest predicted tide for the year above the highest predicted tide for the year, given by $o_c(I_{pc}(j,k),j,k) - p_c(I_{pc}(j,k),j,k)$ for *civil days* and $o_d(I_{pd}(j,k),j,k) - p_d(I_{pd}(j,k),j,k)$ for *daylight days*. The *third metric* is essentially a measure of the residual, or storm surge, on the day of the highest predicted tide for the year.

6. Finally, the three *annual metrics* (5(a) to 5(c), above) were averaged over all valid years for each *record* (these are here called *averaged metrics*) and presented on global maps in Figs 1 to 6. The spread of the first two *metrics* (5(a) and 5(b), above) over the valid years are presented as complementary cumulative distribution functions (CCDFs; otherwise called 'exceedance distributions') in Figs 7 to 9.

This resulted in three types of *annual* and *averaged metrics* for each *record*, and for each of the two kinds of 'day' (*civil days* and *daylight days*).

It should be noted that the results presented here are based on comparisons of the observed sea level with tidal predictions derived from a two-year period of observations which include the time of the observation. Therefore, the results are only indicative of intra-annual (e.g. seasonal) deviations of observations from predictions, rather than of inter-annual deviations (e.g. those due to the El Niño-Southern Oscillation) or long-term trends (e.g, sea-level rise). Inclusion of these latter effects would have required the selection of longer, and therefore fewer, tide-gauge records. Such effects would be expected expand the regions where *WKT* would not perform well.

Tidal analysis and prediction broadly followed Cartwright (1985), with the tidal analysis using singular value decomposition (Press et al., 2007) for the least-squares solution.

## 3 Results

### 3.1 The *averaged first metric*

Figs. 1 and 2 show the *averaged first metric* for *civil days* and *daylight days*, respectively. The Figures indicate how much higher, on average, the annual maximum observed sea level is above the maximum observed on the day of the highest predicted tide for the year (the *WKT Day*); in other words, how much better it would have been if the *WKT* photography had been done on the day of the annual maximum observed sea level rather than on the *WKT Day* (these days only rarely coincide, as discussed in Section 3.4 and shown in Figs. 7 to 9). As we might expect, Figs. 1 and 2 show that there is little obvious difference between the results for *civil days* and *daylight days*. The same is true for the other two *metrics* (Figs. 3 to 6) and, therefore, for Section 3.4 only the results for *daylight days* are shown.





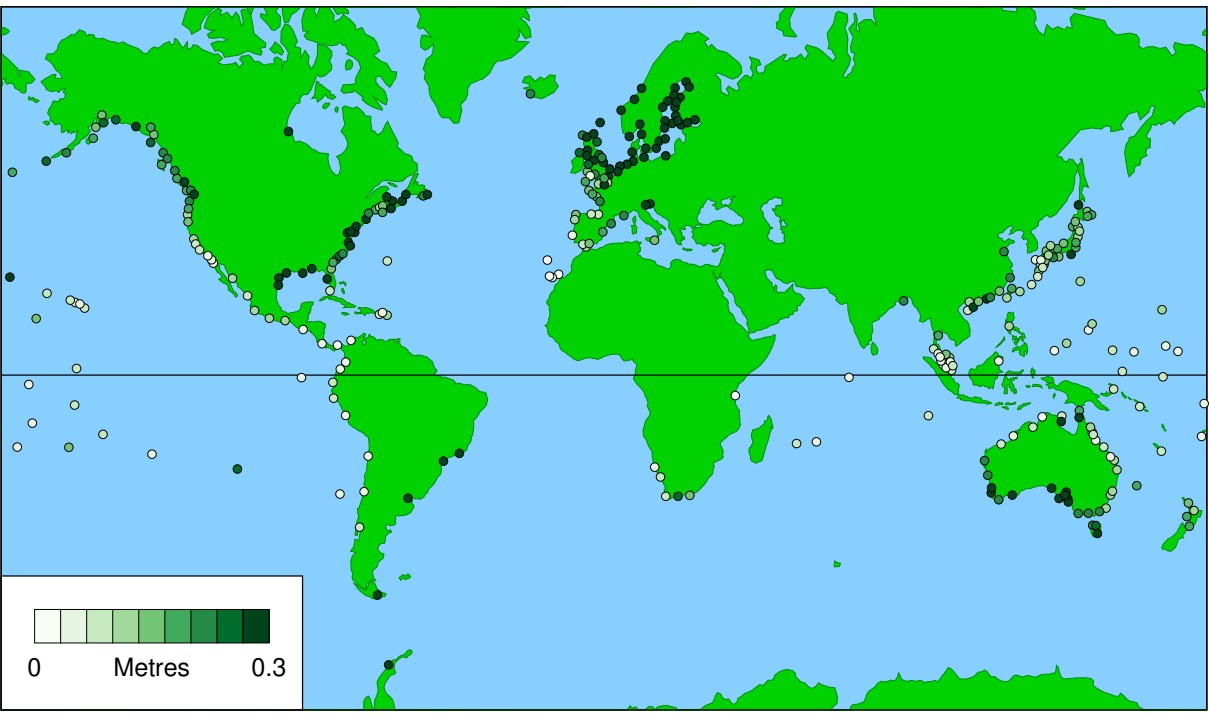

**Figure 1.** *Averaged first metric*, which is height of highest observed sea level above observed maximum on day of highest predicted tide for the year, for *civil days* ($o_c(I_{oc}(j,k),j,k) - o_c(I_{pc}(j,k),j,k)$), averaged over all valid years, $j$.

Fig. 2 (for *daylight days*) provides a guide to where in the world *WKT* is likely to be successful (low values, light colour) and where it is not (high values, dark green). The large white and dark green circles show the locations of the *records* discussed in Section 3.4 and the white and dark green ellipses show the regions discussed in Section 4.

## 3.2 The *averaged second metric*

Figs. 3 and 4 show the *averaged second metric* for *civil days* and *daylight days*, respectively. The Figures indicate the number of days during the year when the sea level was higher than it was on the day of the highest predicted tide for the year (the *WKT Day*); in other words, how many other better opportunities there were during the year for *WKT* photography than on the *WKT Day*.

Again, the results for *civil days* and *daylight days* are very similar. Figure 4 (for *daylight days*) provides another guide to where in the world *WKT* is likely to be successful (low values, light colour) and where it is not (high values, dark green).



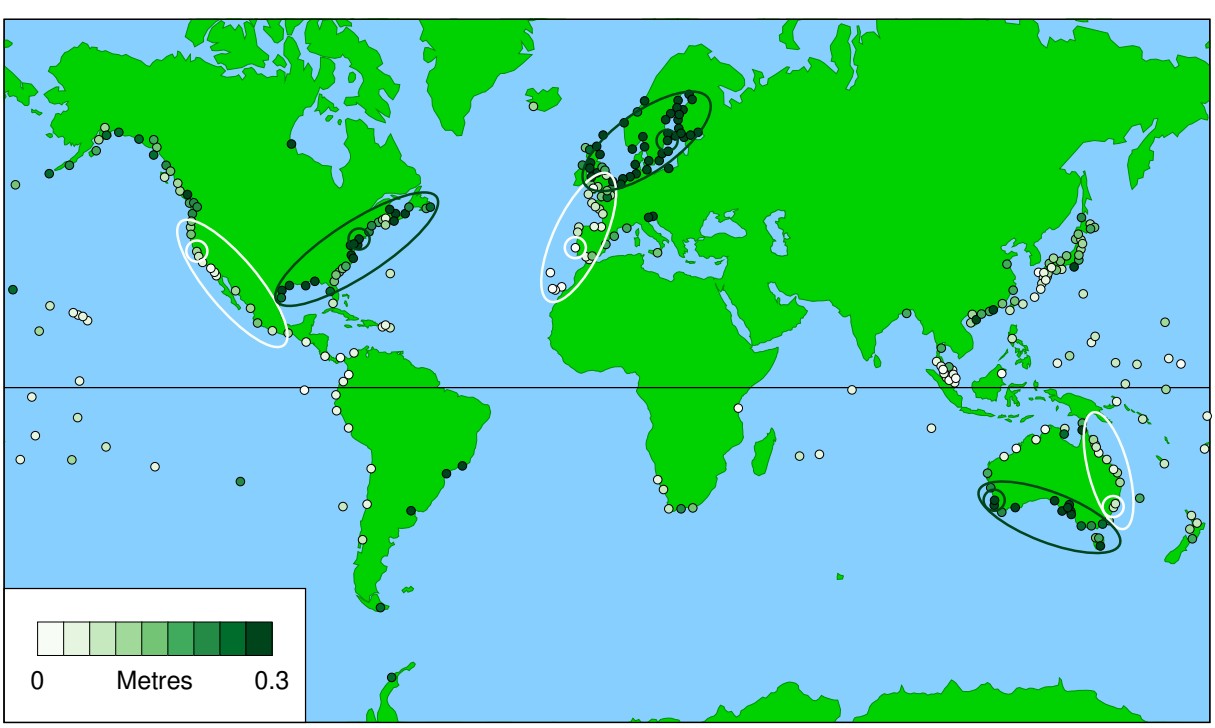

**Figure 2.** *Averaged first metric*, which is height of highest observed sea level above observed maximum on day of highest predicted tide for the year, for *daylight days* ($o_d(I_{od}(j,k),j,k) - o_d(I_{pd}(j,k),j,k)$), averaged over all valid years, $j$. The large white and dark green circles indicate the *records* for the results shown in Figs. 7, 8 and 9. The white and dark green ellipses indicate the regions discussed in Section 4.





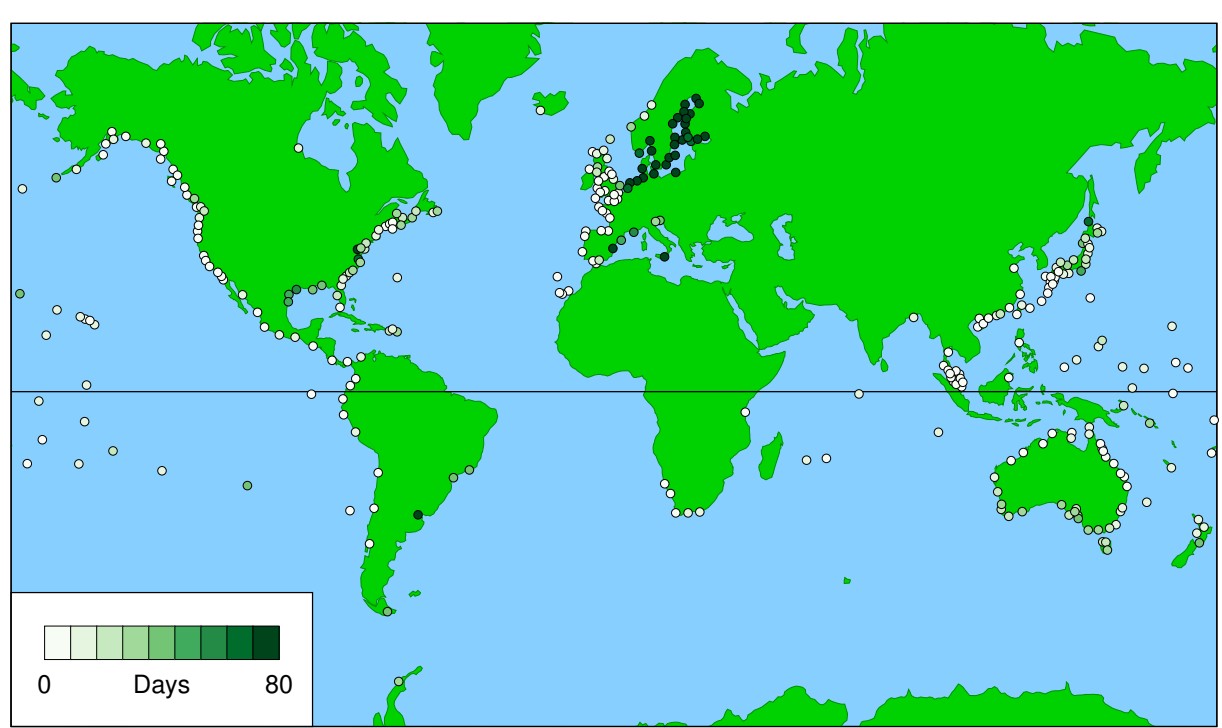

**Figure 3.** *Averaged second metric*, which is no. of days when observed sea level, $o_c(i,j,k)$, higher than observed maximum on day of highest predicted tide for the year, $o_c(I_{pc}(j,k),j,k)$, for *civil days*, averaged over all valid years, $j$.



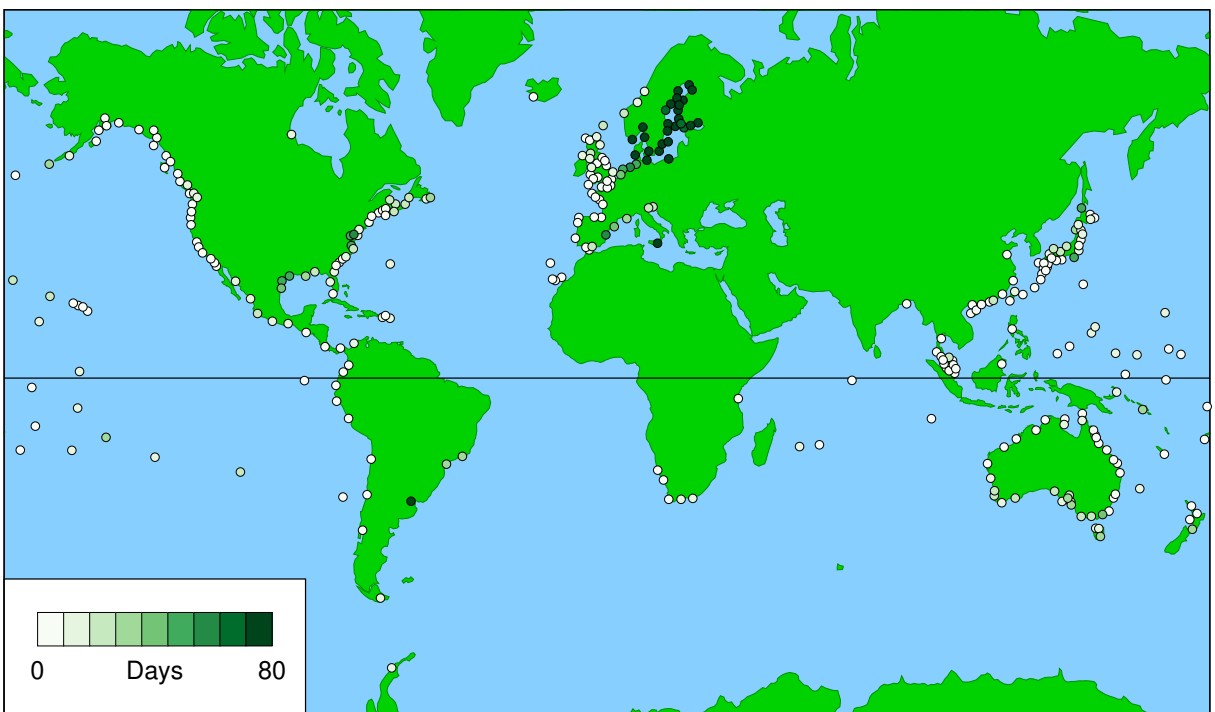

**Figure 4.** *Averaged second metric*, which is no. of days when observed sea level, $o_d(i, j, k)$, higher than observed maximum on day of highest predicted tide for the year, $o_d(I_{pd}(j, k), j, k)$, for *daylight days*, averaged over all valid years, $j$.

### 3.3  The *averaged third metric*

Figs. 5 and 6 show the *averaged third metric* for *civil days* and *daylight days*, respectively. The Figures show the difference between the highest observed and predicted sea levels on the day of the highest predicted tide for the year, which is essentially a measure of the residual, or storm surge, on that day. This *metric* can have either sign (for positive of negative surges).

5     Again, the results for *civil days* and *daylight days* are very similar. Fig. 6 (for *daylight days*) provides another guide to the usefulness of *WKT* in various parts of the world. However, in this case, the *metric* operates in the opposite direction to the other two. In cases where it is negative (light colour), the negative surge would clearly be problematic for *WKT* whereas, in cases where it is positive (dark green), the positive surge would be a bonus.



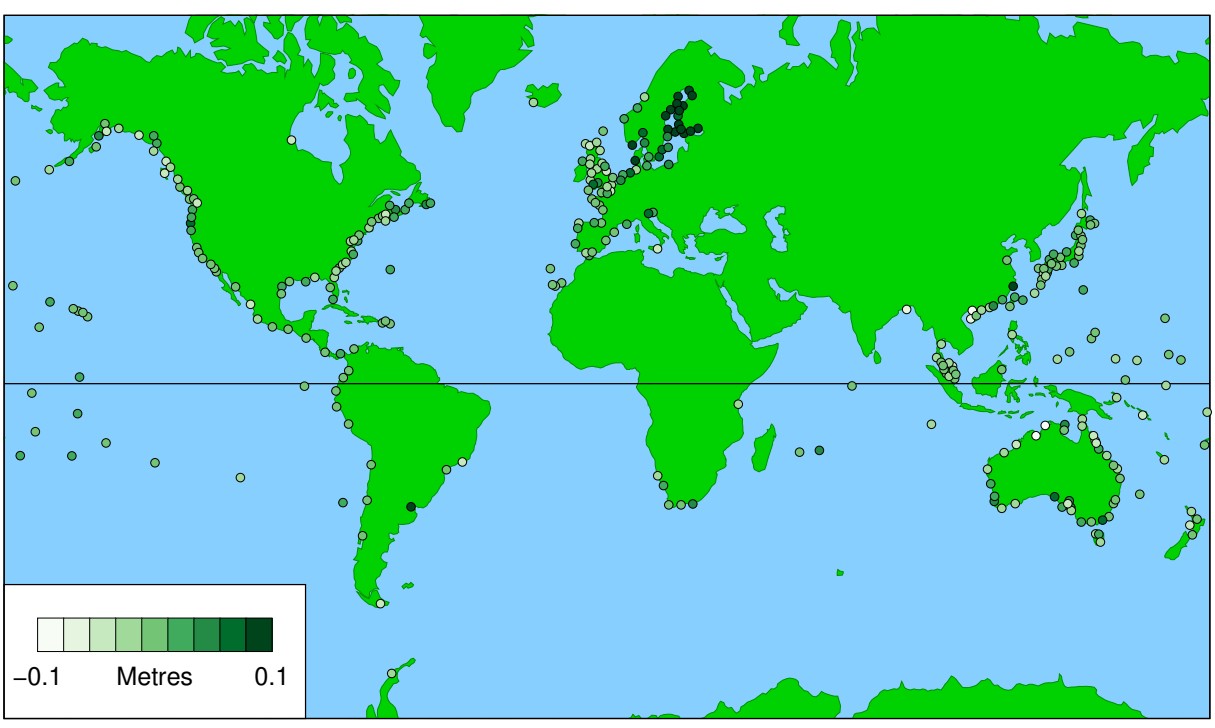

**Figure 5.** *Averaged third metric*, which is height of highest observed sea level on day of highest predicted tide for the year above highest predicted tide for the year, for *civil days* $(o_c(I_{pc}(j,k),j,k) - p_c(I_{pc}(j,k),j,k))$, averaged over all valid years, $j$.

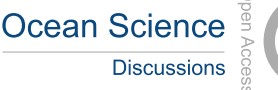

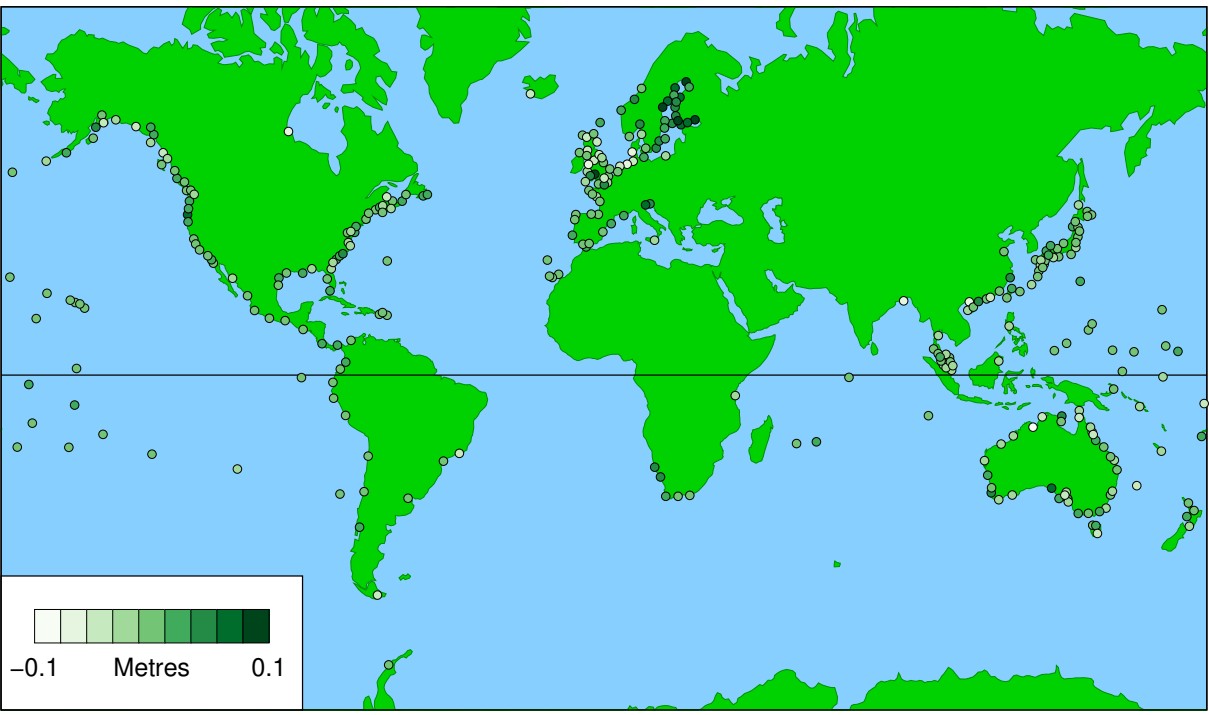

**Figure 6.** *Averaged third metric*, which is height of highest observed sea level on day of highest predicted tide for the year above highest predicted tide for the year, for *daylight days* ($o_d(I_{pd}(j,k),j,k) - p_d(I_{pd}(j,k),j,k)$), averaged over all valid years, $j$.

### 3.4 The distribution of the *annual first* and *second metrics* for six typical locations on three continents

Sections 3.1 to 3.3 discuss three *averaged metrics* derived from 311 *records* which have records that contained at least 20 valid years of data and which have been 'pruned' from the original 586 *records* for display on a global map. Here are presented the complementary cumulative distribution functions (CCDFs; otherwise called 'exceedance distributions') of the *annual first* and

5 *second metrics* for *daylight days* for all valid years of data for six *records* on three continents (San Francisco and New York in the USA; Cascais (near Lisbon) and Stockholm in Europe; Fremantle and Fort Denison (Sydney) in Australia). The locations have been selected because they illustrate, within each continent, very different fitness for *WKT*. The average and median values of these *annual metrics* for *daylight days* (i.e. those shown in Figs. 2 and 4) are shown in Table 1. The differences in fitness for *WKT* is evident from the significant differences of these values within each pair.

10 Two things should initially be noted about Figs. 7 to 9:

1. The intercepts on the vertical (CCDF) axes for any one location are the same for the *first* and *second metrics*. This is because years for which the *annual first metric* is zero are the same as the years for which the *annual second metric* is zero (i.e. when the highest sea-level of the year occurs on the *WKT Day*).





**Table 1.** First column: location. Second column: no. of valid years in analysis. Third and fourth columns: average and median of *annual first metric*, which is height of highest observed sea level above observed maximum on day of highest predicted tide for the year, for *daylight days* $(o_d(I_{od}(j,k),j,k) - o_d(I_{pd}(j,k),j,k))$, over all valid years, $j$. Fifth and sixth columns: average and median of *annual second metric*, which is no. of days when observed sea level, $o_d(i,j,k)$, higher than observed maximum on day of highest predicted tide for the year, $o_d(I_{pd}(j,k),j,k)$, for *daylight days*, over all valid years, $j$.

| Location | Valid years | Annual first metric (metres) | | Annual second metric (days) | |
|---|---|---|---|---|---|
| | | Average | Median | Average | Median |
| San Francisco | 114 | 0.11 | 0.09 | 4.1 | 2 |
| New York | 66 | 0.40 | 0.38 | 13.6 | 12 |
| Cascais (near Lisbon) | 33 | 0.03 | 0.00 | 1.4 | 0 |
| Stockholm | 119 | 0.35 | 0.35 | 84.2 | 57 |
| Fremantle | 93 | 0.30 | 0.30 | 21.5 | 17 |
| Fort Denison (Sydney) | 95 | 0.07 | 0.05 | 4.3 | 2 |

2. The pairs of CCDFs all overlap to a certain extent. Therefore, although one site may perform better on average than the other site, there are always some years at the first site that are worse than some years at the other site. A measure of this overlap may be provided by the proportion of *annual metric* values for one site that falls within the full range of *annual metric* values for the other site; this is discussed for each pair of sites in the following sections.

### 3.4.1  San Francisco and New York

Fig. 7 shows the CCDFs of the *annual first* and *second metrics* for *daylight days* for San Francisco (one of the large white circles in Fig. 2) and New York (one of the large green circles in Fig. 2) in the USA. The CCDFs for both *annual metrics* are significantly narrower for San Francisco (averages of 0.11 m and 4.1 days, respectively; see Table 1) than for New York (averages of 0.40 m and 13.6 days, respectively). On this basis, San Francisco seems a better candidate for *WKT* than New York.

However, there is considerable variability from year to year and considerable overlap of the CCDFs for the two sites. 51% of the *annual first metrics* at New York falls within the full range of the *annual first metrics* at San Francisco, while 86% of the *annual second metrics* at New York falls within the full range of the *annual second metrics* at San Francisco.

### 3.4.2  Cascais (near Lisbon) and Stockholm

Fig. 8 shows the CCDFs of the *annual first* and *second metrics* for *daylight days* for Cascais (near Lisbon; one of the large white circles in Fig. 2) and Stockholm (one of the large green circles in Fig. 2) in Europe. The CCDFs for both *annual metrics* are significantly narrower for Cascais (averages of 0.03 m and 1.4 days, respectively; see Table 1) than for Stockholm (averages of 0.35 m and 84.2 days, respectively). Cascais is clearly a better candidate for *WKT* than Stockholm.





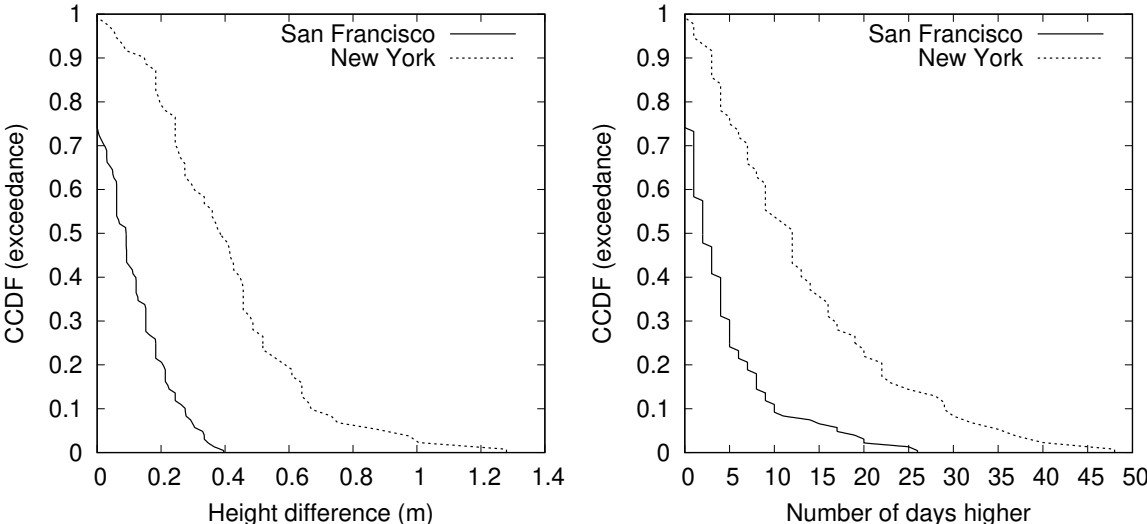

**Figure 7.** Complementary cumulative distribution functions (CCDFs) for San Francisco and New York. Left panel: *annual first metric*, which is height of highest observed sea level above observed maximum on day of highest predicted tide for the year, for *daylight days* $(o_d(I_{od}(j,k),j,k) - o_d(I_{pd}(j,k),j,k))$, estimated over all valid years, $j$. Right panel: *annual second metric*, which is no. of days when observed sea level, $o_d(i,j,k)$, higher than observed maximum on day of highest predicted tide for the year, $o_d(I_{pd}(j,k),j,k)$, for *daylight days*, estimated over all valid years, $j$.

The contrast between Cascais and Stockholm is more marked than for the other two pairs of *records*, Cascais showing very narrow CCDFs, with 50% of the *annual first* and *second metrics* being zero, meaning that the highest sea-level of the year occurred on the *WKT Day*. Only 13% of the *annual first metrics* at Stockholm falls within the full range of the *annual first metrics* at Cascais, while only 7% of the *annual second metrics* at Stockholm falls within the full range of the *annual second metrics* at Cascais. Cascais is clearly a good candidate for WKT.

### 3.4.3 Fremantle and Fort Denison (Sydney)

Fig. 9 shows the CCDFs of the *annual first* and *second metrics* for *daylight days* for Fremantle (one of the large green circles in Fig. 2) and Fort Denison (Sydney; one of the large white circles in Fig. 2) in Australia.

Qualitatively, the relationship between Fremantle and Fort Denison is similar to that between New York and San Francisco, Fort Denison and San Francisco being the better candidates for WKT. The CCDFs for both *annual metrics* are significantly narrower for Fort Denison (averages of 0.07 m and 4.3 days, respectively; see Table 1) than for Fremantle (averages of 0.30 m and 21.5 days, respectively).



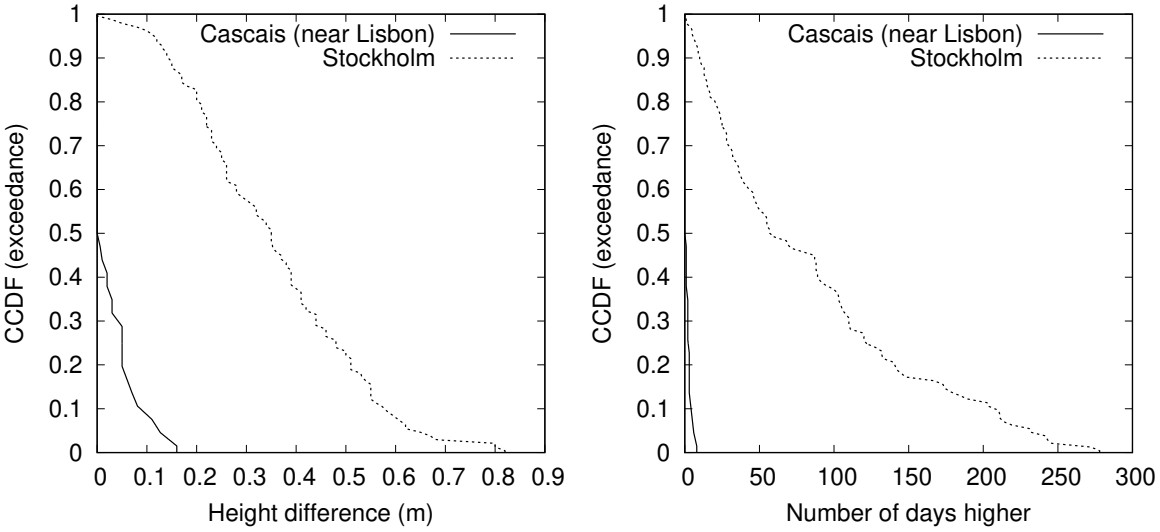

**Figure 8.** Complementary cumulative distribution functions (CCDFs) for Cascais (near Lisbon) and Stockholm. Left panel: *annual first metric*, which is height of highest observed sea level above observed maximum on day of highest predicted tide for the year, for *daylight days* $(o_d(I_{od}(j,k),j,k) - o_d(I_{pd}(j,k),j,k))$, estimated over all valid years, $j$. Right panel: *annual second metric*, which is no. of days when observed sea level, $o_d(i,j,k)$, higher than observed maximum on day of highest predicted tide for the year, $o_d(I_{pd}(j,k),j,k)$, for *daylight days*, estimated over all valid years, $j$.

Again, there is considerable variability from year to year and considerable overlap of the CCDFs for the two sites. 56% of the *annual first metrics* at Fremantle falls within the full range of the *annual first metrics* at Fort Denison. 58% of the *annual second metrics* at Fremantle falls within the full range of the *annual second metrics* at Fort Denison

### 3.5 The variances of the observed sea level and of the predicted tide

5  As noted in the Introduction, the success of *WKT* depends strongly on the size of the storm surge (which is indicated by the *third metric*, displayed in Figs. 5 and 6) relative to the tide; in general, strong storm surges confound attempts to predict the day when *WKT* would be successful while, if the storm surge were always zero, the *WKT Day* (i.e. the day with the highest predicted tide of the year) would always be the day of the highest sea level of the year. It is therefore possible that the relative magnitudes of storm surge and tide could provide a simple alternative to the *metrics* discussed earlier. Fig. 10 shows the ratio

10  of the variance of the observed sea level to the variance of the predicted tide (both calculated in the same way as for the derivation of the *metrics*, as described in the Methods section). It provides another guide to where in the world *WKT* is likely to be successful (low values, light colour) and where it is not (high values, dark green).



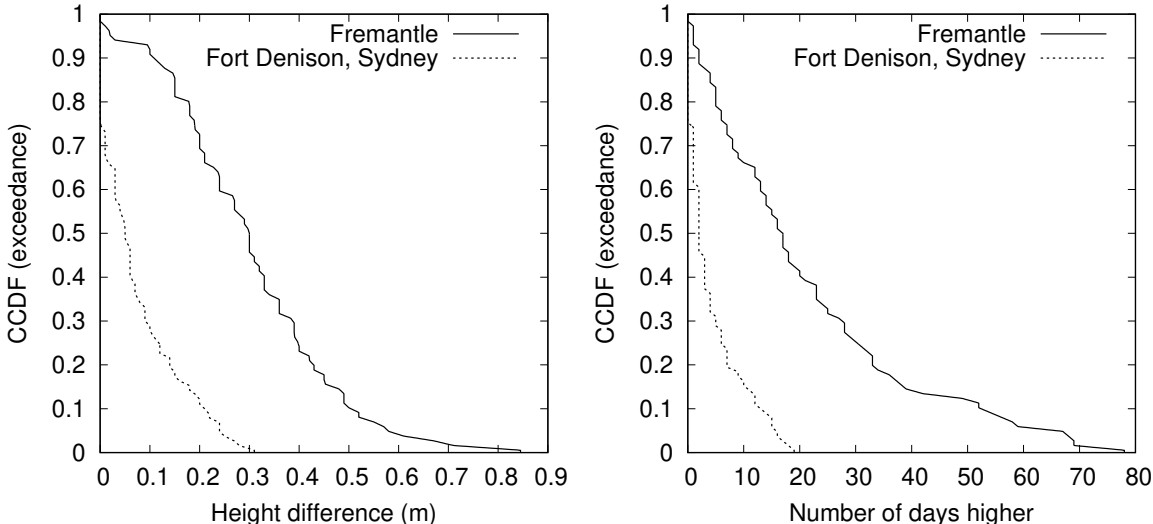

**Figure 9.** Complementary cumulative distribution functions (CCDFs) for Fremantle and Fort Denison (Sydney). Left panel: *annual first metric*, which is height of highest observed sea level above observed maximum on day of highest predicted tide for the year, for *daylight days* ($o_d(I_{od}(j,k),j,k) - o_d(I_{pd}(j,k),j,k)$), estimated over all valid years, $j$. Right panel: *annual second metric*, which is no. of days when observed sea level, $o_d(i,j,k)$, higher than observed maximum on day of highest predicted tide for the year, $o_d(I_{pd}(j,k),j,k)$, for *daylight days*, estimated over all valid years, $j$.

## 4   Discussion

Figs. 1 to 6 provide maps showing the three *metrics*, averaged over at least 20 valid years for 311 tide-gauge *records*. The best measures for suitability for *WKT* are the *averaged first* and *second metrics* (Figs. 1 to 4), as they are based on observations throughout each of the years analysed. Sites where it would be expected that *WKT* would perform well are indicated by low

5   values (light colour), while high values (dark green) suggest poor performance.

Less useful, though nevertheless interesting, is the *third metric* (Figs. 5 to 6), which shows the storm surge averaged over all *WKT Days*; it is less useful than the other *metrics* because it is based solely on information from *WKT Days*. In cases where it is negative (light colour), the negative surge would clearly be problematic for *WKT* whereas, in cases where it is positive (dark green), the positive surge would be a bonus.

10   Figs. 1 to 6 are presented in two ways: for *civil days* (i.e. the normal 24-hour day) and *daylight days* (i.e. the periods over which a natural-light photo may reasonably be taken). Inspection of the Figures indicates that there is little difference between the results for *civil days* and *daylight days*, and so the following discussion relates only to the results for *daylight days*.

Fig. 2 (the *averaged first metric* for *daylight days*) indicates three regions where *WKT* should perform well (white ellipses):

–   the west coast of the USA,





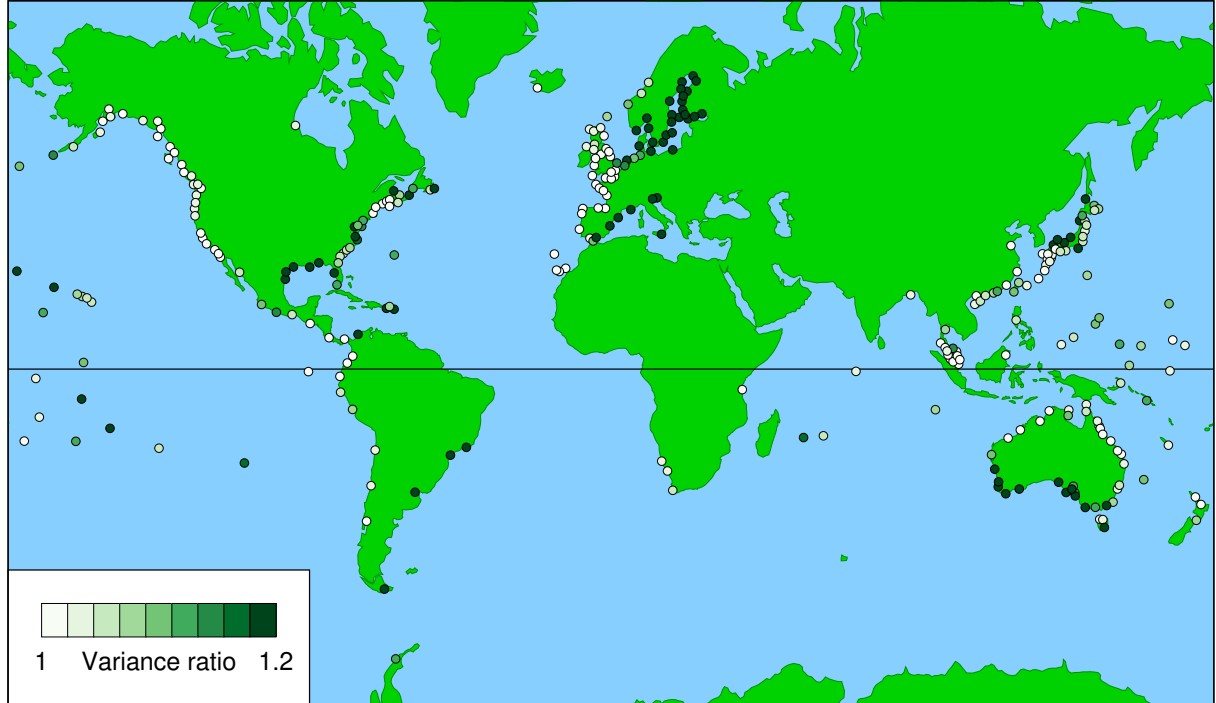

**Figure 10.** Ratio of variance of observed sea level to variance of predicted tide.

- southwestern Europe and locations off northwestern Africa, and

- the east coast of the Australian mainland,

and three regions where *WKT* should perform poorly (dark green ellipses):

- the east coast of North America,

5    - northern Europe, and

- the south and southwest coast of the Australian mainland and Tasmania.

These regions coincide with the pairs of typical *records* shown in Figs. 7 to 9 and summarised in Table 1. It appears fortuitous that the first *WKT* project was conducted in New South Wales, which is the region around Fort Denison (Sydney), shown by the white circle in southeastern Australia in Fig. 2. The large values of the *averaged first metric* in northern Europe are related

10   to a combination of weak tides (e.g. in the Baltic; Stigebrandt, 2001) and significant surges (e.g. in the North Sea; Huthnance, 1991).

Fig. 4 (the *averaged second metric* for *daylight days*) shows generally the same features as Fig. 2 but the contrasts are not so marked. Low values in southwestern Europe and locations off northwestern Africa, and high values in Northern Europe are clear, but the variations in North America and Australia are more subtle.


Figure 10 shows an alternative estimator of the viability of WKT, which is the ratio of the variance of the observed sea level to the variance of the predicted tide (again derived from *records* with at least 20 years of valid data); in this case, *WKT* is likely to be viable at sites with a low value (light colour). Figure 10 shows many of the features displayed by the *first metric* for *daylight days* (Fig 2), indicating that this simple estimator may be as useful as the *first metric* in determining regional variations

in the the performance of *WKT*.

## 5   Conclusions

Figs. 2 and 10 provide useful preliminary indicators of regions where a *WKT* project may be successful, in the sense that the day of highest predicted tide for the year (the *WKT Day*) would yield an observed level comparable with the maximum observed level for the year. However, it is suggested that, prior to initiating a *WKT* project, local tide-gauge records that are

longer than 20 years are analysed in ways similar to those described here (e.g. the production of figures similar to Figs. 7 to 9) to provide a more detailed assessment of the viability of *WKT*.

It is, however, unclear whether the *WKT* strategy (i.e. picking, in advance, the day when the coast is to be photographed) is the best one. An attractive alternative is to photograph every high tide of the year and pick, in retrospect, the images which show the highest sea level. This procedure could be quite easily performed using the camera of a smartphone, suitably programmed

to take photos at the required times and to transmit them to a central repository.

*Data availability.* Tide-gauge data used in these analyses was obtained from the database, Global Extreme Sea Level Analysis Version 2 (GESLA-2): https://gesla.org/, accessed: 11 March 2016 and 19 March 2016.

## Appendix A: The method of pruning records into 'neighbourhoods'

In order to reduce the density of the locations of *records*, the locations were divided into groups which are here called *neigh-*

*bourhoods*. A *neighbourhood* is a unique and objectively defined group of locations in which every location is within a prescribed distance, $d$, of at least one other location in that *neighbourhood*. In a similar way to houses in a neighbourhood, a house is close to one or more of its neighbours, but not necessarily close to all the other houses in the neighbourhood. The method proceeds as follows:

1. Calculate symmetric $n \times n$ matrix $A_{i,j}$ of spheroidal distances between all $n$ locations.

2. For all $(i,j)$, if $A_{i,j} > d$ set $A_{i,j} = 0$, otherwise set $A_{i,j} = 1$, where $d$ is a prescribed distance. An entry of '1' in $A_{i,j}$ therefore indicates that the pair of locations are 'close'.

3. Matrix multiply $A_{i,j}$ with itself to yield another symmetric matrix, $B_{i,j}$ (i.e. $B_{i,j} = A_{i,j}^2$), and set all finite values of B to 1 (i.e. if $B_{i,j} > 0$ then $B_{i,j} = 1$).





4. If $A_{i,j} \neq B_{i,j}$, set $A_{i,j}$ to $B_{i,j}$ and go to 3, otherwise finish.

The resultant matrix, $B_{i,j}$, generally contains numerous repeated rows (and columns, because $B_{i,j}$ is symmetric). $B_{i,j}$ may be simplified by removing any rows that are repeated, yielding a non-symmetric $m \times n$ matrix, $C_{i,j}$ where $m$ is the number of *neighbourhoods*. $C_{i,j}$ represents a table indicating in which *neighbourhood* a given location lies (the $j$th location lies in the $i$th

5  *neighbourhood*, if $C_{i,j} = 1$). Each column of $C_{i,j}$ contain a single '1', because a location can only lie in one *neighbourhood*.

The above procedure converges quickly. For $d = 75 km$, the locations of the 586 *records* yielded 311 *neighbourhoods* and required only 4 iterations of steps (3) and (4),

*Competing interests.* The author declares that he has no conflict of interest.

*Acknowledgements.* I am very grateful to Philip Woodworth for discussions and collaborations concerning tides (e.g. the development of the
10  GESLA-1 and GESLA-2 databases) over many years.





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
