# Peer review of "Are tidal predictions a good guide to future extremes? - a critique of the Witness King Tides Project"

_Ocean Science, 2019_

## Referee Comment (RC1) · Phil Watson (Referee) · 13 Feb 2020

From the outset, I have no concern over the quite detailed analysis undertaken and presented within the manuscript which for all intents and purposes provides an interesting insight into the difference between the highest predicted astronomical tide during any given year and the actual highest recorded water levels around the world's coastlines.

However, as a critique on the utility of the actual "Witness King Tides Project", I have concerns that the modest objectives of the original projects in Australia have been unwittingly mispresented in the manuscript as to confer a more measurable, scientific output from these citizen science endeavours.

As noted correctly by the author, the idea had its origins in January 2009 in NSW,

[Figure]

Australia but the objectives of the exercise were indeed quite modest from a scientific perspective. The intention first and foremost was to use the predictable coincidence of a king tide visible during daylight hours, to raise public awareness at the more fundamental local level about the prospect of predicted sea level rise from climate change and how that might impact local landscapes using a king tide as a visual reference plane of sorts. In its crudest form, the public messaging was as simple as visualising water levels possibly up to a metre deeper than what you observe of the king tide by the end of the century under high range sea level projections.

The day needed to be set well in advance to have the opportunity to condition the public and align state and local government staff participation in what proved a stunningly successful public awareness initiative that has grown roots and expanded rapidly with more accessible internet and telecommunications tools. The event itself was augmented by numerous publications, presentations and media to explain the relationships between predicted tides, actual water levels and sea level projections into the future. The initiative quickly evolved into a national and more recently international event.

The paper makes the point at line 13 that "A critical assumption of WKT is that the annual highest astronomical tide is a good proxy for the actual highest water level during the year, both in timing and height" and then goes on to scientifically address this assumption. However, this as described in the objectives outlined above, was never a critical assumption in developing the concept. A high predicted tide, visible during daylight hours, with sufficient time to promote, provide technical support and public messaging, coordinate and attend to relevant IT requirements were all key considerations in planning such an undertaking. Despite technological advancements and well-established modern WKT networks, engaging with the public in a meaningful way through these citizen science style projects still requires planning and commitments in advance that wont necessarily line up with real-time physical phenomena that can significantly raise water levels above predicted tides as noted by the author.

In the main, my key concern is that the objectives of the WKT are not accurately described in the paper. The linkages to scientific assessment of the difference between the highest predicted tide in any year and the peak measured water level during daylight hours are almost incongruous to critiquing the WKT project? Some thoughts perhaps for the author to consider.

Thankyou for the opportunity to review the work.

———————————————

---

## Short Comment (SC1) · 9 Mar 2020

Overall, I believe that this study contains many useful insights into the causes of high sea levels and associated coastal inundation and their spatial variability around the world. In particular, the spatial distribution of where annual maximum sea levels is tide-dominated compared to surge-dominated helps to explain spatial variability in extreme sea level drivers and projections around the world. These are important findings and make this paper an important contribution to the coastal inundation literature. However, I feel that the definition of 'success' of Witness King Tides has been too narrowly defined, especially when viewed from the perspective as a mechanism to generate coastal inundation impact information.

This study has largely ignored one of the key motivations of Witness King Tides – to document coastal inundation impacts that will occur increasingly frequently with sea-level rise. Rather, it has rather narrowly considered the single question of whether high sea levels coincided with high predicted tides on some of the days that images were taken during the project. For example, the appropriateness of using a single 'WKT day' per year (as opposed to once-a-month or once-a-decade, for example) in the metrics defined has not been discussed. The existence or value of coastal assets that are impacted by king tides (e.g. Hanslow et al. 2018) also not been considered in the assessment of the suitability of sites for WKT locations. These are both important factors to consider when assessing the success of a coastal monitoring program. For example, recent research by Hague et al. (2019) used WKT and other sources (e.g. social media, online news) to show that there is large spatial variability in coastal inundation frequency across Australia. In some places coastal inundation was reported many times per year and in others it occurred less frequently than one year. The reasons for these spatial differences are likely many, but a lack of coastal infrastructure built close to the high tide marks was noted at some locations where coastal inundation occurred infrequently.

This leads into my key point – that just because the highest annual sea level didn't coincide with the highest predicted tide it doesn't make WKT 'not successful'. WKT is one of few coastal change monitoring programs – two other notable cases from Australia are Fluker Posts (Augar and Fluker 2015) and CoastSnap (Harley et al. 2019). The reduction in activity in WKT in the last 5 years has resulted in a large reduction of reports of coastal inundation impacts (e.g. refer Witness King Tides' Flickr page: https://www.flickr.com/photos/witnesskingtides/). To my knowledge this has not been replaced by an alternative publicly-available source – impact reports are now primarily confined to private or institutional repositories, portions of which are occasionally published in reports or research studies (e.g. Maddox 2018, Hague et al. 2019). Unlocking, collating, or generating, these sources of impacts information is vital to understand the physical impacts of coastal inundation and how the frequency and nature

of these will change as sea-level rise continues and accelerates. The continuation and enhancements of programs such as WKT are of great scientific importance. Concerningly, the results of this study could be (mis)interpreted to suggest that we only need to monitor coastal impacts at locations where inundation is tide-dominated. This is a dangerous proposition when coastal inundation impact information is simultaneously becoming rarer but also more important as scientists consider the impacts of sea-level rise on coastal communities happening now and in the future.

The 'attractive alternative' offered by the author – to photograph every high tide and pick the ones associated with the highest sea level – is effectively advocating for CoastSnap or Fluker Posts to be extended to, or expanded in, areas where coastal inundation occurs. (CoastSnap is currently confined to open ocean environments where coastal erosion is being monitored.) This is an excellent idea, and one that would likely be successful, with enough financial or community support for the project. However, other new technologies such as flying of drones (Klemas 2015) or use of social media analytics (Hino et al. 2019) could also provide opportunities for citizen science coastal monitoring projects and should also be considered as alternatives. I would however suggest that the aim of future programs is to capture any day where coastal inundation occurs, rather than the highest annual sea level. This will ensure a focus on coastal inundation impacts, rather than simply extreme sea levels.

Finally, regarding the results discussed in Section 3.5 and shown in Figure 10 – that the ratio of variances of observed sea levels and predicted tides may be a simpler but suitable metric for assessing the relative proportions of tide-dominated and surge-dominated extreme sea level regimes. It would have been interesting to further explore whether tidal range is a key factor in this analysis. For example, are low ratios due to infrequent storm surges or because tidal range is large? This could be useful to investigate due to its implications in the changing predictability of coastal inundation and potentially highlight locations where increases in tide-dominated inundation are most pronounced, and hence help identify candidate locations for future monitoring

efforts.

References

Augar, N. and Fluker, M. (2015) Towards Understanding User Perceptions of a Tourist-based Environmental Monitoring System: An Exploratory Case Study, Asia Pacific Journal of Tourism Research, 20, 10, 1081-1093, DOI: 10.1080/10941665.2014.962554

Hague, B. S, Murphy, B. F., Jones, D. A. and Taylor, A. J. (2019) Developing impact-based thresholds for coastal inundation from tide gauge observations. Journal of Southern Hemisphere Earth System Science. DOI: 10.22499/3.6902.006 [accepted: http://www.bom.gov.au/jshess/docs/2019/Hague2_early.pdf]

Hanslow, D. J., Morris, B. D., Foulsham, E. and Kinsela, M. A. (2018) A Regional Scale Approach to Assessing Current and Potential Future Exposure to Tidal Inundation in Different Types of Estuaries. Sci. Rep., 8, DOI:10.1038/s41598-018-25410-y.

Harley, M.D., Kinsela, M.A., Sanchez-Garcia, E. and Vos, K. (2019) Shoreline change mapping using crowd-sourced smartphone images, Coastal Engineering, 150, 175-189.

Hino, M., Belanger, S. T., Field, C. B., Davies, A. R. and Mach, K. J. (2019) High-tide flooding disrupts local economic activity. Sci. Adv., 5, DOI:eaau2736.

Klemas, V. V. (2015) Coastal and Environmental Remote Sensing from Unmanned Aerial Vehicles: An Overview. Journal of Coastal Research, 31, 5, 1260 – 1267.

Maddox, S. (2018) NSW Ocean and River Entrance Tidal Levels Annual Summary 2017-2018. Manly Hydraulics Laboratory Report MHL2618, State of New South Wales, Sydney.

---

## Referee Comment (RC2) · Ivan Haigh (Referee) · 10 Mar 2020

In this paper John Hunter uses records from a quasi-global tide-gauge dataset to determine in which regions of the world the Witness King Tide (WKT) Project would perform well and other regions where it would not perform well. Overall, I find the paper to be interesting, novel and well written – and commend the author for a nice study. The statistical approach is robust. Therefore, I recommend it for publication. However, I have a few moderate/minor corrections that I feel should be undertaken to strengthen the paper.

Moderate Comments

I think it might be useful, in the introduction, to include a short paragraph describing why

the predicted tidal height changes through the year and maybe even include a figure showing a year of tidal predictions at a semi-diurnal and diurnal example site. I still find people don't appreciate or understand the differences in height and timing in a given year between semi-diurnal, diurnal or mixed tidal sites. For example, you could mention that the largest semidiurnal tidal range occurs in March and September during the equinoxes, while the largest diurnal tidal range occurs in June and December during the solstices. The day of largest tide varies with phasing of the spring and neap tidal cycle and influence of moons distance to earth (e.g., perigee).

I wondered whether it would be interesting, for one year or a couple of years, to plot the actual date when the maximum tide occurs, as this will vary quite a bit around the world depending on whether the site has semi-diurnal, diurnal or mixed tides.

No where do you mention the 4.4 and 18.6 tidal cycles – these can be important in influencing both the timing and height of the annual maximum predicted tide from year to year, but also in a given year. I assume these are accounted for in the tidal analysis.

In the paper there is no mention of storm-surges induced by tropical cyclones. These can be very large. I was just wondering how such events might influence/bias the results around the tropics.

I don't feel too strongly about this, but I wonder whether a short paragraph, or few sentences could be added to briefly highlight the papers that have looked at sunny day or nuisance flooding, as this has some relevance here.

Minor Comments

Page 1, Line 11 – you could add that this has become known as 'Sunny day flooding' nor 'nuisance flooding'.

Page 1, line 6 – maybe add a sentence or two to describe why there might be more than one astronomical tides of similar magnitude to the maximum. For example, larger than average tides occur twice per year around either the equinoxes or solstices depending

on whether you have semi-diurnal or diurnal tides.

Page 2, line 3 – I would replace 'and tidal observations' with 'and sea level observations' as you are considering both tide and surge.

Page 2, line 14 – some justification is need for the first selection. How much data was ignored based on this selection.

Page 2, line 15 – I am not sure what you mean by 'binned' – do you mean averaged or interpolated.

Page 3, line 5 — which tidal analysis software was used?

Sorry for my delay in posting this review.

---

## Author Comment (AC2) · 20 Mar 2020

I thank Ben Hague for his comments and am gratified that he feels that the manuscript is "an important contribution to the coastal inundation literature". However, he claims that "the definition of 'success' of Witness King Tides has been too narrowly defined, especially when viewed from the perspective as a mechanism to generate coastal inundation impact information". I would argue that the manuscript does not attempt to define the "success" of Witness King Tides (WKT) but rather to indicate places where the observed sea level on the "WKT Day" is unlikely to be "unusually high" (see below for my discussion of "unusually high") with the result that none of the stated objectives could be properly met. I do not attempt to discuss any other attributes which could contribute to the "success" of a WKT project, nor suggest any other ways in which a

[Figure]

WKT project could be deemed "unsuccessful".

There are two readily-accessible definitions of the purposes of Witness King Tides, firstly from the website www.kingtides.net:

"We are citizen scientists, capturing data and images showing what the future sea levels will be and what is at risk. The King Tides Project helps people all over the world understand how sea level rise will impact their lives.

King Tides photos are used several ways to help people:

1. Document current flood risk in coastal areas. 2. Visualize the impacts of future sea level rise in their community. 3. Ground-truth and validate climate change models by comparing model predictions with the high-tide reality. 4. Serve as a living record of change for future generations."

Secondly, from the report of the first WKT project by Watson and Fraser (Watson and Fraser, 2009; reference in main manuscript), which defined the two "primary objectives" as:

* identifying areas vulnerable to tidal inundation, capturing the tide level against revetments, seawalls, jetties and other marine infrastructure; and

* raising awareness throughout the wider community about the current projections for sea level rise to the end of the century (approximately 90 cm)."

It is clear from the above that the success of a single WKT project requires (among other things) that the maximum sea level on a "WKT day" is unusually high. Given that WKT projects are generally only carried out once per year in any one location, I imply by "unusually high" that the maximum sea level on a "WKT day" is among the highest for the year. WKT makes the assumption that the day of highest predicted tide of the year is a good proxy for a day when the observed sea level is "unusually high" - the manuscript questions this assumption and indicates places where it is probably valid and places where it is not. This is the primary aim of the paper.

The reviewer indicates numerous aspects of WKT that I do not discuss (or intend to discuss) in the manuscript, for example:

(1) "..... the appropriateness of using a single 'WKT day' per year (as opposed to once-a-month or once-a-decade, for example) in the metrics defined has not been discussed."

It isn't discussed because the aim of the manuscript is not to redesign WKT. At least in Australia, this is the way in which WKT started out (in most location, there was one "WKT Day" per year). Unfortunately, the history of WKT both in Australia and globally has been quite poorly documented and it is difficult to get an overall picture of what projects have actually occurred, where and when.

(2) "The existence or value of coastal assets that are impacted by king tides (e.g. Hanslow et al. 2018) (have) also not been considered in the assessment of the suitability of sites for WKT locations."

Again, consideration of coastal assets was not an aim of the manuscript.

(3) "These are both important factors to consider when assessing the success of a coastal monitoring program."

The reviewer appears to believe that the main aim of the paper is to provide a comprehensive assessment of WKT. It is not. It is to investigate one critical requirement for a WKT to have some hope of success - as noted above, it is that the maximum sea level on a "WKT day" is "unusually high". If this requirement is not met (and the manuscript indicates likely places where this might be so) then it can be reasonably argued that the WKT project will fail, in that the resultant images become no more useful than images of random high tides. Indeed, the manuscript warns that such cases could well negate the stated aims of WKT of "raising awareness ..... about the current projections for sea level rise" and to "visualize the impacts of future sea level rise" (see above), by noting that "a significant negative storm surge on a WKT Day may well give the unintended

message that the impact of sea-level rise is likely to be unimportant" (page 1, lines 22-23).

(4) "Concerningly, the results of this study could be (mis)interpreted to suggest that we only need to monitor coastal impacts at locations where inundation is tide-dominated."

The manuscript indicates "regions where a WKT project may be successful, in the sense that the day of highest predicted tide for the year (the WKT Day) would yield an observed level comparable with the maximum observed level for the year" (page 16, lines 7-9). Essentially, WKT works well in tide-dominated regions and poorly in regions not dominated by the tide. It seems a strange leap of logic to suggest that "we only need to monitor coastal impacts at locations where inundation is tide-dominated" just because WKT only works well in these regions. The manuscript even suggests (page 16, lines 13-15) an alternative strategy to WKT for possible use in regions where WKT may not work well.

(5) "I would however suggest that the aim of future programs is to capture any day where coastal inundation occurs, rather than the highest annual sea level. This will ensure a focus on coastal inundation impacts, rather than simply extreme sea levels."

This is exactly what the suggested alternative strategy (page 16, lines 13-15) would do.

(6) "It would have been interesting to further explore whether tidal range is a key factor in this analysis. For example, are low ratios due to infrequent storm surges or because tidal range is large? This could be useful to investigate due to its implications in the changing predictability of coastal inundation and potentially highlight locations where increases in tide-dominated inundation are most pronounced, and hence help identify candidate locations for future monitoring efforts."

Yes - it could be useful but does not, alas, fall within the scope or aims of this manuscript.

---

## Author Comment (AC3) · 23 Mar 2020

I thank Ivan Haigh for his thoughtful review in which he makes a number of good suggestions for improving the manuscript.

1. "..... you could mention that the largest semidiurnal tidal range occurs in March and September during the equinoxes, while the largest diurnal tidal range occurs in June and December during the solstices. The day of largest tide varies with phasing of the spring and neap tidal cycle and influence of moons distance to earth (e.g., perigee)."

Good idea - this will be included - see point (7), below.

2. "I wondered whether it would be interesting, for one year or a couple of years, to plot the actual date when the maximum tide occurs, as this will vary quite a bit around the

world depending on whether the site has semi-diurnal, diurnal or mixed tides."

I tend to think that the manuscript already has quite enough plots and complexity, and that this information would not really help anyone to assess the feasibility of performing Witness King Tides (WKT) at a given site. As noted in the Conclusions, such preliminary information can be obtained from Figs 2 to 10 of the manuscript and "it is suggested that, prior to initiating a WKT project, local tide-gauge records that are longer than 20 years are analysed in ways similar to those described here ..... to provide a more detailed assessment of the viability of WKT".

3. "Nowhere do you mention the 4.4 and 18.6 tidal cycles - these can be important in influencing both the timing and height of the annual maximum predicted tide from year to year, but also in a given year. I assume these are accounted for in the tidal analysis."

Firstly, I will add an Appendix note describing briefly the lineage of the tidal analysis program. It is based on astronomical arguments and tidal frequencies generated by software provided by the (then) Proudman Oceanographic Laboratory (now the National Oceanography Centre, Liverpool), the techniques described by Cartwright (1985;Tidal Prediction and Modern Time Scales, International Hydrographic Review, Vol. 62, No. 1, 127-138) and singular value decomposition for the linear regression solution.

Secondly, yes - the 4.4 and 18.6 tidal cycles are accounted for, in two ways: (a) tidal modulations are accounted for by "nodal" corrections, calculated by the routines to be discussed in the Appendix (see above), and (b) the tidal analysis (for 102 constituents) is performed on a two-year time series centred on the middle of the year being analysed. This therefore represents a "running" analysis, which would give a reasonable representation of the tidal modulations, even if "nodal" corrections (a) weren't applied. A "running" analysis is used in order to remove interannual variability and to minimise the effects of unidentified vertical datum shifts in the records.

4. "In the paper there is no mention of storm-surges induced by tropical cyclones. These can be very large. I was just wondering how such events might influence/bias

the results around the tropics."

There has been no attempt to separate surges induced by tropical cyclones from other surges (e.g. those induced by mid-latitude synoptic systems). The main problem with surges from tropical cyclones is that there are relatively infrequent and probably under-sampled in some of the records (all of which contained at least 20 valid years). This could possibly contribute to the uncertainty in the results at some locations, but further analysis of this effect is beyond the scope of the manuscript, which is to provide preliminary guidance as to the feasibility of a WKT project.

5. "I don't feel too strongly about this, but I wonder whether a short paragraph, or few sentences could be added to briefly highlight the papers that have looked at sunny day or nuisance flooding, as this has some relevance here."

I don't like the term "sunny day flooding" as it infers that "nuisance flooding" only relates to periods when the storm surge is small compared with the tide (i.e. that it occurs on "sunny days", rather than during storms). However, the term "nuisance flooding" describes the effect (flooding which is of low level, and which only causes minor rather than major disruption or property damage), rather than the cause; therefore it can exist in surge-dominated environments as well as tidally-dominated ones. I have dealt with this issue in the next item (6).

6. "Page 1, Line 11 - you could add that this has become known as "Sunny day flooding" or "nuisance flooding".

Agreed (but I will not use the term "sunny day flooding" for the reason given above): I will add, at the end of the sentence "WKT is a citizen-science project designed to collect photos of the shoreline at the time of annual highest astronomical tide, with the aim of indicating the flooding that may occur routinely with sea-level rise" (page 1, lines 9-11), the reference "(e.g. Moftakhari et al., 2015)", after which I'll insert the sentence:

"Such flooding, if it is of low level and only causes minor rather than major disruption

or property damage, is generally referred to as "nuisance flooding" (Moftakhari et al., 2018)."

The two references are:

Moftakhari, H. R., AghaKouchak, A., Sanders, B. F., Allaire, M., & Matthew, R. A. What is nuisance flooding? Defining and monitoring an emerging challenge, Water Resources Research, 54, 4218-4227, https://doi.org/10.1029/2018WR022828, 2018.

Moftakhari, H. R., A. AghaKouchak, B. F. Sanders, D. L. Feldman, W. Sweet, R. A. Matthew, and A. Luke, Increased nuisance flooding along the coasts of the United States due to sea level rise: Past and future, Geophysical Research Letters, 42, 9846-9852, https://doi.org/10.1002/2015GL066072, 2015.

7. "Page 1, line 6 - maybe add a sentence or two to describe why there might be more than one astronomical tides of similar magnitude to the maximum. For example, larger than average tides occur twice per year around either the equinoxes or solstices depending on whether you have semi-diurnal or diurnal tides."

I'll add, on page 2, line 6, after the sentence ending "of similar to the maximum", the sentence: "If the tides are predominantly semidiurnal, the largest maxima occur near the equinoxes (March and September) and, if the tides are predominantly diurnal, the largest maxima occur near the solstices (June and December); for example, see Ray and Merrifield (2019).". The reference is:

Ray, R. D. and Merrifield, M. A. The semiannual and 4.4-year modulations of extreme high tides, Journal of Geophysical Research: Oceans, 124, 5907-5922, https://doi.org/10.1029/2019JC015061, 2019.

8. "Page 2, line 3 - I would replace 'and (in) tidal observations' with 'and (in) sea level observations' as you are considering both tide and surge."

Thank you - of course, I meant "sea-level observations" - I will change it.

[Figure]

9. "Page 2, line 14 - some justification is need for the first selection. How much data was ignored based on this selection."

After "observed heights that departed by more than 10 standard deviations from the average were rejected" I will insert "(this is a simple check to remove extreme outliers; in the entire GESLA-2 data set of over 300 million data points, only 190 values were rejected in this way)".

10. Page 2, line 15 - I am not sure what you mean by "binned" - do you mean averaged or interpolated.

I mean averaged into bins (e.g. see en.wikipedia.org/wiki/Data_binning). I will change "binned" to "averaged into bins".

11. Page 3, line 5 - which tidal analysis software was used?

See item (3), above.

---

## Author Response (AR1)

Response to Reviews of:'Are tidal predictions a good guide to future extremes?a critique of the Witness King Tides Project',submitted to Ocean Science

**1 Note to the Editors**

While Ivan Haigh provided a useful technical review of the manuscript, I was quite disappointed with the reviews of Phil Watson and Ben Hague. These latter two reviews did not address the technical details of my manuscript at all, but rather mounted strident defences of the Witness King Tides project and made only broad recommendations, which would effectively result in a very different manuscript with very different aims. I have therefore responded to these generalities but found no suggestions that would warrant modification of my manuscript. Luckily, the manuscript was also reviewed by a colleague who gave a number of detailed technical recommendations, most of which have been incorporated. I have likewise addressed the helpful technical comments of Ivan Haigh, the majority of which have also been incorporated.

My responses, below, are in red. Line numbers refer to the original manuscript.

**2 Reviewer: Ivan Haigh**

**2.1 The Full Review**

In this paper John Hunter uses records from a quasi-global tide-gauge dataset to determine in which regions of the world the Witness King Tide (WKT) Project would perform well and other regions where it would not perform well. Overall, I find the paper to be interesting, novel and well written - and commend the author for a nice study. The statistical approach is robust. Therefore, I recommend it for publication. However, I have a few moderate/minor corrections that I feel should be undertaken to strengthen the paper.

**Moderate Comments**

I think it might be useful, in the introduction, to include a short paragraph describing why the predicted tidal height changes through the year and maybe even include a figure showing a year of tidal predictions at a semi-diurnal and diurnal example site. I still find people don't appreciate or understand the differences in height and timing in a given year between semi-diurnal, diurnal or mixed tidal sites. For example, you could mention that the largest semidiurnal tidal range occurs in March and September during the equinoxes, while the largest diurnal tidal range occurs in June and December during the solstices. The day of largest tide varies with phasing of the spring and neap tidal cycle and influence of moons distance to earth (e.g., perigee).

I wondered whether it would be interesting, for one year or a couple of years, to plot the actual date when the maximum tide occurs, as this will vary quite a bit around the world

depending on whether the site has semi-diurnal, diurnal or mixed tides.

No where do you mention the 4.4 and 18.6 tidal cycles - these can be important in influencing both the timing and height of the annual maximum predicted tide from year to year, but also in a given year. I assume these are accounted for in the tidal analysis.

In the paper there is no mention of storm-surges induced by tropical cyclones. These can be very large. I was just wondering how such events might influence/bias the results around the tropics.

I don't feel too strongly about this, but I wonder whether a short paragraph, or few sentences could be added to briefly highlight the papers that have looked at sunny day or nuisance flooding, as this has some relevance here.

Minor Comments

Page 1, Line 11 - you could add that this has become known as 'Sunny day flooding' nor 'nuisance flooding'.

Page 1, line 6 - maybe add a sentence or two to describe why there might be more than one astronomical tides of similar magnitude to the maximum. For example, larger than average tides occur twice per year around either the equinoxes or solstices depending on whether you have semi-diurnal or diurnal tides.

Page 2, line 3 - I would replace 'and tidal observations' with 'and sea level observations' as you are considering both tide and surge.

Page 2, line 14 - some justification is need for the first selection. How much data was ignored based on this selection.

Page 2, line 15 - I am not sure what you mean by 'binned' - do you mean averaged or interpolated.

Page 3, line 5 - which tidal analysis software was used? Sorry for my delay in posting this review.

**2.2 The Response**

I thank Ivan Haigh for his thoughtful review in which he makes a number of useful suggestions for improving the manuscript.

1. '... you could mention that the largest semidiurnal tidal range occurs in March and September during the equinoxes, while the largest diurnal tidal range occurs in June and December during the solstices. The day of largest tide varies with phasing of the spring and neap tidal cycle and influence of moons distance to earth (e.g., perigee).' (Page C2, paragraph 1.):

Good idea - this has been addressed - see point (7), below.

2. 'I wondered whether it would be interesting, for one year or a couple of years, to plot the actual date when the maximum tide occurs, as this will vary quite a bit around the world depending on whether the site has semi-diurnal, diurnal or mixed tides.' (Page C2, paragraph 2.):

I tend to think that the manuscript already has quite enough plots and complexity, and that this information would not really help anyone to assess the feasibility of performing Witness King Tides (WKT) at a given site. As noted in the Conclusions, such preliminary information can be obtained from Figs. 2 to 10 of the manuscript and 'it is suggested that, prior to initiating a WKT project, local tide-gauge records that are longer than 20 years are analysed in ways similar to those described here ... to provide a more detailed assessment of the viability of WKT'.

3. 'Nowhere do you mention the 4.4 and 18.6 tidal cycles - these can be important in influencing both the timing and height of the annual maximum predicted tide from year to year, but also in a given year. I assume these are accounted for in the tidal analysis.' (Page C2, paragraph 3.):

Firstly, I have expanded the final paragraph of Section 2 with the addition of a final sentence: 'Astronomical arguments and tidal frequencies were generated by software provided by the (then) Proudman Oceanographic Laboratory (now the National Oceanography Centre, Liverpool, U.K.)'.

Secondly, yes - the 4.4 and 18.6 tidal cycles are accounted for, in two ways: (a) tidal modulations are accounted for by 'nodal' corrections, calculated by the routines described above, and (b) the tidal analysis (for 102 constituents) is performed on a two-year time series centred on the middle of the year being analysed. This therefore represents a 'running' analysis, which would give a reasonable representation of the tidal modulations, even if 'nodal' corrections (a) weren't applied. A 'running' analysis is used in order to remove interannual variability and to minimise the effects of unidentified vertical datum shifts in the records. A sentence to this effect has been included in the previous paragraph to the one that describes the tidal analysis: 'This removes signals of period longer than about two years, and most of the effects of any vertical datum shifts in the tide-gauge records'.

4. 'In the paper there is no mention of storm-surges induced by tropical cyclones. These can be very large. I was just wondering how such events might influence/bias the results around the tropics.'

There has been no attempt to separate surges induced by tropical cyclones from other surges (e.g. those induced by mid-latitude synoptic systems). The main problem with surges from tropical cyclones is that there are relatively infrequent and probably under-sampled in some of the records (all of which contained at least 20 valid years). This could possibly contribute to the uncertainty in the results at some locations, but further analysis of this effect is beyond the scope of the manuscript, which is to provide preliminary guidance as to the feasibility of a WKT project.

5. 'I don't feel too strongly about this, but I wonder whether a short paragraph, or few sentences could be added to briefly highlight the papers that have looked at sunny day or nuisance flooding, as this has some relevance here.'

I don't like the term 'sunny day flooding' as it infers that 'nuisance flooding' only relates to periods when the storm surge is small compared with the tide (i.e. that it occurs on 'sunny days', rather than during storms). However, the term 'nuisance flooding' describes the effect (flooding which is of low level, and which only causes minor rather than major disruption or property damage), rather than the cause; therefore it can exist in surge-dominated environments as well as tidally-dominated ones. I have dealt with this issue in the next item (6).

6. 'Page 1, Line 11 - you could add that this has become known as "Sunny day flooding" or "nuisance flooding".

I agree (but I will not use the term 'sunny day flooding' for the reason given above): I have added, at the end of the sentence 'WKT is a citizen-science project designed to collect photos of the shoreline at the time of annual highest astronomical tide, with the aim of indicating the flooding that may occur routinely with sea-level rise' (page 1, lines 9-11), the reference '(e.g. Moftakhari et al., 2015)', followed by the sentence:

'Such flooding, if it is of low level and only causes minor rather than major disruption or property damage, is generally referred to as "nuisance flooding" (Moftakhari et al., 2018).'

7. 'Page 1, line 6 - maybe add a sentence or two to describe why there might be more than one astronomical tides of similar magnitude to the maximum. For example, larger than average tides occur twice per year around either the equinoxes or solstices depending on whether you have semi-diurnal or diurnal tides.'

I've added, on page 2, line 6, after the sentence ending 'of similar to the maximum', the sentence:

'If the tides are predominantly semidiurnal, the largest maxima occur near the equinoxes (March and September) and, if the tides are predominantly diurnal, the largest maxima occur near the solstices (June and December); for example, see Ray and Merrifield (2019).'

8. 'Page 2, line 3 - I would replace "and (in) tidal observations" with "and (in) sea level observations" as you are considering both tide and surge.'

Thank you - of course, I meant 'sea-level observations' - I have changed it.

9. 'Page 2, line 14 - some justification is need(ed) for the first selection. How much data was ignored based on this selection.'

After 'observed heights that departed by more than 10 standard deviations from the average were rejected' I have inserted '(this is a simple check to remove extreme outliers; in the entire GESLA-2 data set of over 300 million data points, only 190 values were rejected in this way)'.

10. 'Page 2, line 15 - I am not sure what you mean by "binned" - do you mean averaged or interpolated.'

I mean averaged into bins (e.g. see the Wikipedia article). I have changed 'binned' to 'averaged into bins'.

11. 'Page 3, line 5 - which tidal analysis software was used?'See item (3), above.

**3 Reviewer: Phil Watson**

**3.1 The Full Review**

From the outset, I have no concern over the quite detailed analysis undertaken and presented within the manuscript which for all intents and purposes provides an interesting insight into the difference between the highest predicted astronomical tide during any given year and the actual highest recorded water levels around the world's coastlines.

However, as a critique on the utility of the actual 'Witness King Tides Project', I have concerns that the modest objectives of the original projects in Australia have been unwittingly misrepresented in the manuscript as to confer a more measurable, scientific output from these citizen science endeavours.

As noted correctly by the author, the idea had its origins in January 2009 in NSW, Australia but the objectives of the exercise were indeed quite modest from a scientific perspective. The intention first and foremost was to use the predictable coincidence of a king tide visible during daylight hours, to raise public awareness at the more fundamental local level about the prospect of predicted sea level rise from climate change and how that might impact local landscapes using a king tide as a visual reference plane of sorts. In its crudest form, the public messaging was as simple as visualising water levels possibly up to a metre deeper than what you observe of the king tide by the end of the century under high range sea level projections.

The day needed to be set well in advance to have the opportunity to condition the public and align state and local government staff participation in what proved a stunningly successful public awareness initiative that has grown roots and expanded rapidly with more accessible internet and telecommunications tools. The event itself was augmented by numerous publications, presentations and media to explain the relationships between predicted tides, actual water levels and sea level projections into the future. The initiative quickly evolved into a national and more recently international event.

The paper makes the point at line 13 that 'A critical assumption of WKT is that the annual highest astronomical tide is a good proxy for the actual highest water level during the year, both in timing and height' and then goes on to scientifically address this assumption. However, this as described in the objectives outlined above, was never a critical assumption in developing the concept. A high predicted tide, visible during daylight hours, with sufficient time to promote, provide technical support and public messaging, coordinate and attend to relevant IT requirements were all key considerations in planning such an undertaking. Despite technological advancements and well-established modern WKT networks, engaging with the public in a meaningful way through these citizen science style projects still requires planning and commitments in advance that wont necessarily line up with real-time physical phenomena that can significantly raise water levels above predicted tides as noted by the author.

In the main, my key concern is that the objectives of the WKT are not accurately described in the paper. The linkages to scientific assessment of the difference between the highest predicted tide in any year and the peak measured water level during daylight hours are almost incongruous to critiquing the WKT project? Some thoughts perhaps for the author to consider.

Thank you for the opportunity to review the work.

**3.2 The Response**

I thank Phil Watson for his comments. However, I feel that he has missed the main points of the manuscript, which are not to misrepresent the objectives of the Witness King Tides project by criticising its scientific output. Rather, I state that WKT has 'the aim of indicating the flooding that may occur routinely with sea-level rise' (page 1, lines 10-11) and my purpose is to 'indicate regions of the world where WKT should perform well ... and others where it would not' (Abstract, lines 2-3). I do not criticise WKT for any lack of rigorous or quantifiable scientific output.

As indicated in the Conclusions, I (a) show regions of the world where WKT may be successful 'in the sense that the day of highest predicted tide for the year ... would yield an observed level comparable with the maximum observed level for the year' (Page 16, lines 7-9), (b) suggest that local tide-gauge records should be analysed prior to initiating a WKT project in order to assess its likely success, and (c) suggest an alternative approach which is 'to photograph every high tide of the year and pick, in retrospect, the images which show the highest sea level ... using the camera of a smartphone, suitably programmed to take photos at the required times and to transmit them to a central repository' (Page 16, lines 13-15); this could form the basis of an alternative citizen-science project.

I agree with Phil that the primary motive of WKT is 'to raise public awareness ... about the prospect of predicted sea level rise from climate change and how that might impact local landscapes'. However, my manuscript hopefully serves as a warning that WKT should be planned with care and with due cognisance of a possible unintended consequence - namely, that a strong negative storm surge on the day of a WKT project could yield photos that are so underwhelming as to suggest to the general public that the impact of sea-level will be minimal. This is my biggest fear about WKT projects that are planned without a good understanding of the local regime of tides and storm surges.

**4 Reviewer: Ben Hague**

**4.1 The Full Review**

Overall, I believe that this study contains many useful insights into the causes of high sea levels and associated coastal inundation and their spatial variability around the world. In particular, the spatial distribution of where annual maximum sea levels is tide-dominated compared to surge-dominated helps to explain spatial variability in extreme sea level drivers and projections around the world. These are important findings and make this paper an important contribution to the coastal inundation literature. However, I feel that the definition of 'success' of Witness King Tides has been too narrowly defined, especially when viewed from the perspective as a mechanism to generate coastal inundation impact information. This study has largely ignored one of the key motivations of Witness King Tides - to document coastal inundation impacts that will occur increasingly frequently with sea-level rise. Rather, it has rather narrowly considered the single question of whether high sea levels coincided with high predicted tides on some of the days that images were taken during the project. For example, the appropriateness of using a single 'WKT day' per year (as opposed to once-a-month or once-a-decade, for example) in the metrics defined has not been discussed. The existence or value of coastal assets that are impacted by king tides (e.g. Hanslow et al. 2018) also not been considered in the assessment of the suitability of sites for WKT locations. These are both important factors to consider when assessing the success of a coastal monitoring program. For example, recent research by Hague et al. (2019) used WKT and other sources (e.g. social media, online news) to show that there is large spatial variability in coastal inundation frequency across Australia. In some places coastal inundation was reported many times per year and in others it occurred less frequently than one year. The reasons for these spatial differences are likely many, but a lack of coastal infrastructure built close to the high tide marks was noted at some locations where coastal inundation occurred infrequently.

This leads into my key point - that just because the highest annual sea level didn't coincide with the highest predicted tide it doesn't make WKT 'not successful'. WKT is one of few coastal change monitoring programs - two other notable cases from Australia are Fluker Posts (Augar and Fluker 2015) and CoastSnap (Harley et al. 2019). The reduction in activity in WKT in the last 5 years has resulted in a large reduction of reports of coastal inundation impacts (e.g. refer Witness King Tides' Flickr page: https://www.flickr.com/photos/witnesskingtides/). To my knowledge this has not been replaced by an alternative publicly-available source - impact reports are now primarily confined to private or institutional repositories, portions of which are occasionally published in reports or research studies (e.g. Maddox 2018, Hague et al. 2019). Unlocking, collating, or generating, these sources of impacts information is vital to understand the physical impacts of coastal inundation and how the frequency and nature of these will change as sea-level rise continues and accelerates. The continuation and enhancements of programs such as WKT are of great scientific importance. Concerningly, the results of this study could be (mis)interpreted to suggest that we only need to monitor coastal impacts at locations where inundation is tide-dominated. This is a dangerous proposition when coastal inundation impact information is simultaneously becoming rarer but also more important as scientists consider the impacts of sea-level rise on coastal communities happening now and in the future.

The 'attractive alternative' offered by the author - to photograph every high tide and pick the ones associated with the highest sea level - is effectively advocating for CoastSnap or Fluker Posts to be extended to, or expanded in, areas where coastal inundation occurs. (CoastSnap is currently confined to open ocean environments where coastal erosion is being monitored.) This is an excellent idea, and one that would likely be successful, with enough financial or community support for the project. However, other new technologies such as flying of drones (Klemas 2015) or use of social media analytics (Hino et al. 2019) could also provide opportunities for citizen science coastal monitoring projects and should also be considered as alternatives. I would however suggest that the aim of future programs is to capture any day where coastal inundation occurs, rather than the highest annual sea level. This will ensure a focus on coastal inundation impacts, rather than simply extreme sea levels. Finally, regarding the results discussed in Section 3.5 and shown in Figure 10 - that the ratio of variances of observed sea levels and predicted tides may be a simpler but suitable metric for assessing the relative proportions of tide-dominated and surge-dominated extreme sea level regimes. It would have been interesting to further explore whether tidal range is a key factor in this analysis. For example, are low ratios due to infrequent storm surges or because tidal range is large? This could be useful to investigate due to its implications in the changing predictability of coastal inundation and potentially highlight locations where increases in tide-dominated inundation are most pronounced, and hence help identify candidate locations for future monitoring efforts.

Secondly, from the report of the first WKT project by Watson and Fraser (Watson and Fraser, 2009; reference in main manuscript), which defined the two 'primary objectives' as:

- 'identifying areas vulnerable to tidal inundation, capturing the tide level against revetments, seawalls, jetties and other marine infrastructure; and
- raising awareness throughout the wider community about the current projections for sea level rise to the end of the century (approximately 90 cm).'

It is clear from the above that the success of a single WKT project requires (among other things) that the maximum sea level on a 'WKT day' is unusually high. Given that WKT projects are generally only carried out once per year in any one location, I imply by 'unusually high' that the maximum sea level on a 'WKT day' is among the highest for the year. WKT makes the assumption that the day of highest predicted tide of the year is a good proxy for a day when the observed sea level is 'unusually high' - the manuscript questions this assumption and indicates places where it is probably valid and places where it is not. This is the primary aim of the paper.

The reviewer indicates numerous aspects of WKT that I do not discuss (or intend to discuss) in the manuscript, for example:

1. '... the appropriateness of using a single "WKT day" per year (as opposed to once-a-month or once-a-decade, for example) in the metrics defined has not been discussed.'

This isn't discussed because the aim of the manuscript is not to redesign WKT. At least in Australia, this is the way in which WKT started out (in most location, there was one 'WKT Day' per year). Unfortunately, the history of WKT both in Australia and globally has been quite poorly documented, and it is difficult to get an overall picture of what projects have actually occurred, where and when. 2. 'The existence or value of coastal assets that are impacted by king tides (e.g. Hanslow et al. 2018) (have) also not been considered in the assessment of the suitability of sites for WKT locations.'

Again, consideration of coastal assets was not an aim of the manuscript.

3. 'These are both important factors to consider when assessing the success of a coastal monitoring program.'

The reviewer appears to believe that the main aim of the paper is to provide a comprehensive assessment of WKT. It is not. It is to investigate one critical requirement for a WKT to have some hope of success - as noted above, it is that the maximum sea level on a 'WKT day' is 'unusually high'. If this requirement is not met (and the manuscript indicates likely places where this might be so) then it can be reasonably argued that the WKT project will fail, in that the resultant images become no more useful than images of random high tides. Indeed, the manuscript warns that such cases could well negate the stated aims of WKT of 'raising awareness ... about the current projections for sea level rise' and to 'visualize the impacts of future sea level rise' (see above), by noting that 'a significant negative storm surge on a WKT Day may well give the unintended message that the impact of sea-level rise is likely to be unimportant' (page 1, lines 22-23).

4. 'Concerningly, the results of this study could be (mis)interpreted to suggest that we only need to monitor coastal impacts at locations where inundation is tide-dominated.'

The manuscript indicates 'regions where a WKT project may be successful, in the sense that the day of highest predicted tide for the year (the WKT Day) would yield an observed level comparable with the maximum observed level for the year' (page 16, lines 7-9). Essentially, WKT works well in tide-dominated regions and poorly in regions not dominated by the tide. It seems a strange leap of logic to suggest that 'we only need to monitor coastal impacts at locations where inundation is tide-dominated' just because WKT only works well in these regions. The manuscript even suggests (page 16, lines 13-15) an alternative strategy to WKT for possible use in regions where WKT may not work well.

5. 'I would however suggest that the aim of future programs is to capture any day where coastal inundation occurs, rather than the highest annual sea level. This will ensure a focus on coastal inundation impacts, rather than simply extreme sea levels.'

This is exactly what the suggested alternative strategy (page 16, lines 13-15) would do.

6. 'It would have been interesting to further explore whether tidal range is a key factor in this analysis. For example, are low ratios due to infrequent storm surges or because tidal range is large? This could be useful to investigate due to its implications in the changing predictability of coastal inundation and potentially highlight locations where increases in tide-dominated inundation are most pronounced, and hence help identify candidate locations for future monitoring efforts.'

Yes - it could be useful but does not, alas, fall within the scope or aims of this manuscript.

John Hunter, 8 April 2020

[revised manuscript text omitted]

---

## Author Response (AR2)

Response to Topic Editor Regarding:'Are tidal predictions a good guide to future extremes?a critique of the Witness King Tides Project',submitted to Ocean Science

**1 Comments to Author**

A few editorial points, as I try to bridge the gap between author and reviewers.

First, Ben Hague should not be criticized for lack of a thorough review, since he was not an official reviewer. The official reviewers are marked by 'RC: Referee comment' on the OS webpage. (Yes, the OS web page does state: 'SC1: Referee review of...'. I have no idea why since he was NOT a referee!) During 'open season' the journal allows anyone to comment on a paper, and Hague's was in this category. He is free to comment on as much or as little as he wishes. He did feel strongly enough about his points to make a substantial comment, complete with a number of useful references. I don't agree with all his points – in fact, he is quite mistaken to say 'this study could be (mis)interpreted to suggest that we only need to monitor coastal impacts at locations where inundation is tide-dominated.' I don't see any danger at all that someone could make that misinterpretation.

Nonetheless, both Watson and Hague clearly think the WKT project is being unfairly criticized. My own take is that much of this rests on the definition of 'success'. Both of them agree that a WKT Day can turn out having lower than predicted sea level. (Who could disagree?) The paper implies the WKT project is a failure, or at least not a 'success', if/when this occurs.

Both Watson and Hague, however, maintain the project has other goals, and given the difficulties with 'citizen science' and outreach on climate issues, I am inclined to grant them some leeway on the definition of success. For example, both claim that one goal was merely to get people to visualize water levels up to a meter higher than they observe on the WKT Day. That can surely be done even if the water level isn't at the highest point for the year.

Therefore, I think that some of these 'soft' goals of the project be mentioned when it is first described, including especially the stress for outreach work on sea level. (For example, visualizing future sea level is listed #2 in the four points of the author-reply to Hague.) A short paragraph at the beginning on these aspects of WKT seems appropriate. Such acknowledgement that the project has wider citizen-science aims will hopefully placate both Watson and Hague. After that, one can move on to the problem of inferring too much from a single day's observations. This should also be done briefly in the Abstract, 'where WKT should perform well' can be rephrased in terms of the stricter view of what 'perform well' means – namely truly observing an extreme sea level on the given day.

In the revised version, I think the new sentence at Line 15 (page 1) should be removed. It may well be true, but it sounds unduly critical and really it adds nothing to the present paper. The WKT project's documentation of past work is irrelevant to the topic of the paper.

Very minor: page 8, line 4 – 'positive of'  $\rightarrow$  'positive or'

Also on page 8, the sentence 'In cases where it is negative... bonus' is exactly repeated on page 14 in the Discussion. But perhaps this was done for emphasis?

**2 The Response**

My responses, below, are in red. Line numbers refer to the first revision of the manuscript.

1. Therefore, I think that some of these 'soft' goals of the project be mentioned when it is first described, including especially the stress for outreach work on sea level. (For example, visualizing future sea level is listed #2 in the four points of the author-reply to Hague.) A short paragraph at the beginning on these aspects of WKT seems appropriate. Such acknowledgement that the project has wider citizen-science aims will hopefully placate both Watson and Hague. After that, one can move on to the problem of inferring too much from a single day's observations.

I have changed the second and third sentences of the introduction to read: 'WKT is a citizen-science project designed to raise awareness of the coastal impacts of future sea-level rise, and to visually document the flooding that occurs at times of unusually high sea levels during the year. One of the main activities of WKT is the collection of photographs of the shoreline at the time of annual highest astronomical tide, with the aim of indicating the flooding that may occur routinely with sea-level rise (Moftakhari et al., 2015)'.

I have also inserted the following sentence at the beginning of the second paragraph of the introduction: 'While WKT is a useful way of raising awareness of the possible impacts of a higher sea level, there is, unfortunately, no perfect way of selecting a suitable WKT Day in advance'.

2. This should also be done briefly in the Abstract, 'where WKT should perform well' can be rephrased in terms of the stricter view of what 'perform well' means – namely truly observing an extreme sea level on the given day.

I have modified the second sentence of the abstract to read: 'The results indicate regions of the world where a key criterion for a WKT project (that it be executed on a day of unusually high sea level) would likely be met (e.g. the west coast of the USA) and others where it would not (e.g. the east coast of North America)'.

3. In the revised version, I think the new sentence at Line 15 (page 1) should be removed. It may well be true, but it sounds unduly critical and really it adds nothing to the present paper. The WKT project's documentation of past work is irrelevant to the topic of the paper.

Done.

4. Very minor: page 8, line 4 – 'positive of'  $\rightarrow$  'positive or' Done.

5. Also on page 8, the sentence 'In cases where it is negative... bonus' is exactly repeated on page 14 in the Discussion. But perhaps this was done for emphasis?

This was intentional - the first part of the sentence is the main reason for my doing the work and so I have left it as it is. I really want to emphasise this point and have added the additional clause '*indeed, this was a prime impetus for the present work*' to the second instance of the sentence, which occurs in the Discussion.

I have also made a couple of minor changes to improve the text (the removal of double quotes to 'nuisance flooding' on page 1, and the change of 'spheroidal' to 'ellipsoidal' on page 17).

John Hunter, 2 May 2020

[revised manuscript text omitted]